# Operation Strategy of Parking Lots Integrated with PV and Considering Energy Price Tags

**Zexin Yang [1,*]**, **Xueliang Huang [1]**, **Shan Gao [1]**, **Qi Zhao [2]**, **Hongen Ding [2]**, **Tian Gao [1]**, **Dongyu Mao [1]** and **Rui Ye [1]**

1 School of Electrical Engineering, Southeast University, Nanjing 210096, China; xlhuang_ee@seu.edu.cn (X.H.); gaoshan@seu.edu.cn (S.G.); 220203003@seu.edu.cn (T.G.); maodongyu@seu.edu.cn (D.M.); 230159506@seu.edu.cn (R.Y.)

2 State Grid Jiangsu Electric Power Co., Ltd., Suzhou Power Supply Branch, Suzhou 215004, China; qzhao1989@163.com (Q.Z.); hongending@163.com (H.D.)

* Correspondence: zexin_yang@seu.edu.cn

**Abstract:** In recent years, the orderly charging of electric vehicles (EVs) in commercial parking has become a meaningful research topic due to the increasing number of EVs, especially for parking lots close to workplaces and serving fixed users. In this paper, a parking lot energy management system integrated with energy storage system (ESS) and photovoltaic (PV) system is established. The concept of energy price tag (EPT) is introduced to define the price of all energy storage devices, and the priority order between PV, ESS, EVs, and power grid is established. Taking the minimization of charging cost as the optimization objective, the charging plans of ESS, EVs, and buildings are optimized considering the constraints of EVs user demand and PV power. By comparing the simulation results of four cases, it is proven that this strategy can reduce the charging cost and improve the consumption rate of PV.

**Keywords:** electric vehicles (EVs); energy storage system (ESS); photovoltaic (PV) system; energy price tags (EPT); parking lot energy management system





## 1. Introduction

In recent years, with the continuous deterioration of environmental pollution and energy depletion, all countries have been vigorously developing clean energy. Electric vehicles (EVs) are becoming more and more popular because of their convenience, beauty, quietness, and little environmental pollution. EVs have the potential to reduce fossil energy consumption and greenhouse gas emissions [1–3]. Under an economic arrangement, EVs can cooperate with solar energy devices to improve the consumption rate of solar energy and the reliability of the whole system, increasing the penetration of sustainable energy sources in our daily lives [4,5].

The distributed energy storage characteristics of EVs provide abundant potential schedulable resources for new energy consumption. It is feasible and amenable to install PV systems in large parking lots to provide electricity for EVs [6]. However, the disordered charging of EVs will cause adverse effects on the power grid. Therefore, it is necessary to install ESS to optimize the coordination of EVs and ESS to maximize PV consumption [7–9].

The parking time of EVs in the workplace parking lot is generally greater than the charging time. Using the schedulable potential of EVs to develop a reasonable charging scheme can not only reduce operating costs, but also reduce the impact on the power grid. In [10], a new parking lot coordinated charging model with PV energy and ESS was introduced which considers the impact of charging prices on EVs charging plans. An EVs charging optimization scheduling model considering dynamic cost is proposed in [11]. This method takes into account the charging demand of EVs and the real-time information of PV power generation but does not consider the ESS. In order to solve the problem that the average waiting time of EVs in new energy charging stations is too long, a constrained Markov decision process method is proposed in [12]. Reference [13] also

proposed a heuristic-based charging optimization strategy for EVs and analyzed the impact of ESS capacity on operating costs.

The above literature ignored the energy prices and operation cost when formulating charging and discharging schemes for energy storage devices, and also neglected to thoroughly consider the energy sources and energy prices in energy storage devices. In [14], a new term is introduced, called energy price tag (EPT), which represents the average price of energy storage. Based on the research of [14], this research extends the application of EPT to other energy storage devices, focusing on the energy management of smart charging stations equipped with PV and ESS.

In this paper, a parking lot energy management system integrated with ESS and PV is proposed. Taking the minimization of charging cost as the optimization objective, different energy priority orders and power allocation methods are established based on EPTs, and the charging plans of ESS, EVs, and buildings are reasonably optimized. The effectiveness of the method is proven by four different cases.

## 2. Parking Lots Model Establishment

The smart parking lot system proposed in this paper is shown in Figure 1. The comprehensive parking lot considered in this paper is located in a commercial building or a nearby parking lot, and the building and the parking lot's component PV panels and fixed storage batteries are connected to the same bus. Assume that all EVs support charging mode and discharging mode. The energy generated by the PV system is first provided to the parking lot, but the output power of PV power generation is greatly affected by the environmental weather conditions. When the PV power is insufficient, the power of the ESS or the grid is used to meet the load demand of the parking lot. When the PV power is surplus to requirements, it is reasonable and beneficial to be used to meet the load demand of the building and to store energy instead of directly returning the power to the grid. It would be better if the imbalance between supply and demand can be self-digested.

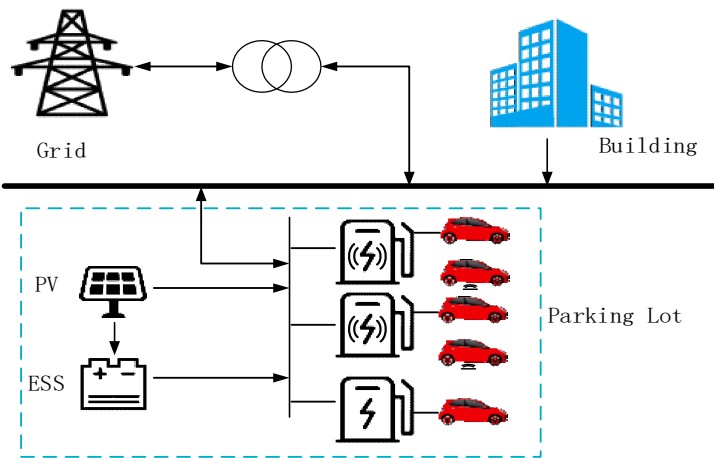

**Figure 1.** The structure of parking lot energy management system.

### 2.1. Proposed Dynamic Pricing for Energy Storage Devices

In [14], the author introduces the average cumulative price of stored energy in EVs, called energy price tag (EPT). According to the EPT concept, any energy storage device can be powered by multiple energy sources, each of which has a fixed price. The EPT calculation formula for each EV is as follows:

$$EPT_{EV}(h) = \begin{cases} \frac{EPT_{EV}^{h-1} \times Energy_{EV}^{h-1} + \sum_{i=1}^{N_s} E_{S-EV}^i \times C_s^i}{Energy_{EV}}, & \text{charging} \\ EPT_{EV}(h-1), & \text{discharging} \end{cases} \quad (1)$$

where $E_{S-EV}^i$ is the energy provided by the energy number $i$ for charging the EV, $C_s^i$ is the price of energy source, Ns is the total number of energy sources, and $EPT_{EV}$ is the energy level of the EV battery after charging.

Accordingly, for the parking lot system shown in Figure 1, PV, battery, and power grid are considered as the energy sources of EVs. Therefore, the instantaneous EPT of each EV charging is calculated:

$$EPT_{EV}(h) = \begin{cases} \frac{EPT_{EV}^{h-1} \times Energy_{EV}^{h-1} + E_{grid-EV} \times C_{grid} + E_{PV-EV} \times C_{pv} + E_{BS-EV} \times C_{BS}}{Energy_{EV}}, & \text{charging} \\ EPT_{EV}(h-1), & \text{discharging} \end{cases} \quad (2)$$

where $E_{grid \to EV}$ is the electricity purchased for EVs from the power grid, $C_{grid}$ is the price of power grid energy. $E_{PV \to EV}$ is the charging energy provided by PV for EVs, and $E_{BS \to EV}$ is the charging energy provided by energy storage for EVs.

The concept of EPT is introduced into the parking station system studied in this paper, the energy and electricity prices of ESS and EVs are defined. When the EPT of the energy storage unit is lower than the grid price, it can be used as a potential energy source to provide energy for other equipment. In this case, the selling price of the energy storage unit should be higher than the EPT and lower than the grid price, so as to ensure that the energy storage unit can obtain income from the sale of electricity. When the EPT of the energy storage unit is higher than the grid price, no electricity is sold, which ensures that the energy used by each unit has the lowest unit price. In order to achieve this goal, the sales price of EVs and energy storage devices is calculated by

$$C_{EV} = \frac{EPT_{EV} + C_{grid}}{2} \quad (3)$$

$$C_{BS} = \frac{EPT_{BS} + C_{grid}}{2} \quad (4)$$

Therefore, EVs and energy storage devices involved in the sale of energy will have different discharging prices, which can calculate the average price of EVs and energy storage devices selling energy:

$$\overline{C_{EV}} = \frac{\sum_{i=1}^I C_{EV}^i \times E_{dis}^i}{\sum_{i=1}^I E_{dis}^i} \quad (5)$$

$$\overline{C_{BS}} = \frac{\sum_{i=1}^I C_{BS}^i \times E_{dis}^i}{\sum_{i=1}^I E_{dis}^i} \quad (6)$$

*2.2. Energy Collaborative Management*

Figure 2 shows different combinations of energy exchange paths between PV parking components.

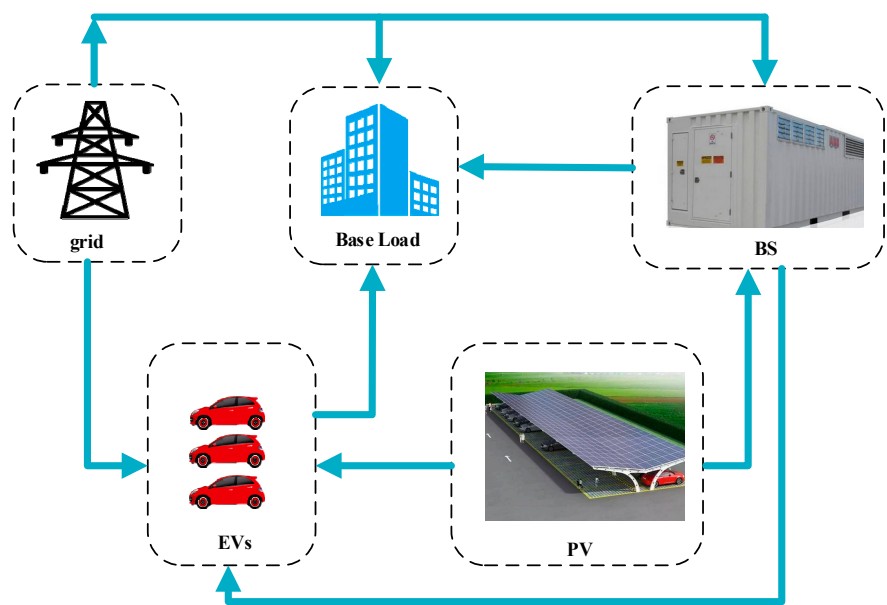

**Figure 2.** Energy path between different components in parking lot.

The energy flow path of PV at each time *t* can be expressed as

$$P_t^{PV} = P_t^{PV-BS} + P_t^{PV-EV} + P_t^{PV-Building} \tag{7}$$

where $P_t^{PV-BS}$, $P_t^{PV-EV}$, and $P_t^{PV-Building}$ respectively represent the charging power provided by PV for BS, EVs, and building.

The energy flow path of energy storage at each time *t* can be expressed as

$$P_t^{BS} = \begin{cases} P_t^{PV-BS} + P_t^{Grid-BS} & \text{charging} \\ P_t^{BS-EV} + P_t^{BS-Building} & \text{discharging} \end{cases} \tag{8}$$

where $P_t^{Grid-BS}$ represents the charging power provided by the grid for EVs; $P_t^{BS-EV}$ is the power from BS, which charges the EV; $P_t^{BS-Building}$ is the charging power provided by the BS for the building.

The energy exchange path of an EV at each time *t* can be expressed as

$$P_t^{EV} = \begin{cases} P_t^{PV-EV} + P_t^{BS-EV} + P_t^{Grid-EV} & \text{charging} \\ P_t^{EV-Building} & \text{discharing} \end{cases} \tag{9}$$

where $P_t^{Grid-EV}$ is the power from grid, which charges the EV; $P_t^{EV-Building}$ is the power from EV, which charges the building.

The energy flow path of the power grid at each time *t* can be expressed as

$$P_t^{Grid} = P_t^{Grid-BS} + P_t^{Grid-EV} + P_t^{Grid-Building} \tag{10}$$

Taking PV charging for EV, PV charging for BS, BS for charging EVs, and EVs discharging as examples, this paper introduces a method for determining priority and energy distribution based on the EPT of energy storage devices.

### 2.2.1. PV Power to Charge EVs

From the above analysis, it can be found that the energy sources of parking lots mainly include: PV installed in parking lot and power grid. The priority order of PV is the highest,

and all EVs in the parking lot have the opportunity to charge using PV energy. The parking lot allocates PV energy according to the EPT of each EV:

$$P_{i,t}^{PV-EV,ch} = P_t^{pv} \times \frac{EPT_i}{\sum_{i=1}^{n} EPT_i} \tag{11}$$

where $EPT_i$ is the energy price tag of $i$th EV.

Considering the actual charging demand of EVs, the final charging power of each EV should meet:

$$P_{i,t}^{PV-EV,ch} = \min(P_{i,t}^{PV-EV,ch}, P_{i,t}^{ch}) \tag{12}$$

### 2.2.2. PV Power to Charge BS

PV output gives priority to providing energy for EVs, and if there is a surplus, it needs to provide energy for energy storage devices.

The power used by PV to charge energy storage is calculated by

$$P_{i,t}^{PV-BS,ch} = \min((P_t^{pv} - P_{cs,t}^{ch}) \times \frac{EPT_i}{\sum_{i=1}^{n} EPT_i}, P_{\max}^{BS,ch}) \tag{13}$$

At this time, if there is more PV surplus, all of it will be provided to the building.

### 2.2.3. Discharge of EVs

When PV output meet EVs electricity and BS electricity requirements, EVs can profit by selling energy to the grid. The parking lot can customize the discharge number of EVs according to the actual situation of EVs in the station:

$$P_{i,t}^{EV-building,dch} = P_{cs,t}^{dch} \times \frac{\frac{1}{EPT_i}}{\sum_{i=1}^{n} \frac{1}{EPT_i}} \tag{14}$$

The discharge capacity of the parking lot is calculated by the following formula:

$$P_{cs,t}^{dch} = \sum_{i=1}^{n} P_{i,t}^{dch} \tag{15}$$

where $n$ represents all the charging piles operating in the parking lot at this time.

### 2.2.4. BS Power to Charge EVs

The PV output cannot meet the charging demand of EVs and BS meets the discharging conditions. According to EPT, power allocation is carried out:

$$P_{i,t}^{BS-EV,ch} = P_t^{BS} \times \frac{\frac{1}{EPT_i}}{\sum_{i=1}^{n} \frac{1}{EPT_i}} \tag{16}$$

### 2.3. Optimization Model

The parking lot system designed in this paper includes PV and ESS. Therefore, when considering the total cost, it is necessary to consider the energy costs purchased from other energy sources and the benefits obtained by selling energy.

Since the energy of PV cannot be guaranteed to meet the charging requirements of all EVs and buildings, it is necessary to buy electricity from the grid. In this case, EVs whose EPT is higher than the grid price are charged first, which can reduce their electricity prices. For buildings, all remaining energy is purchased from the grid.

For parking lots, the energy consumption cost at each moment is defined as:

$$Cost_{station} = \sum_{t=1}^{T} \left\{ P_t^{PV-EV} \times C_{PV}^t + P_t^{PV-Building} \times C_{PV}^t + P_t^{Grid} \times C_{grid}^t + P_t^{BS} \times C_{BS}^t \right\} \tag{17}$$

Correspondingly, the benefit of parking lot at each moment is:

$$Revenue_{station} = -\sum_{t=1}^{T}\left\{ P_t^{EV-Building} \times C_{EV}^t \right\} \tag{18}$$

In summary, the comprehensive cost of parking lot is

$$C_{station} = Cost_{station} + Revenue_{station} \tag{19}$$

- For each EV:
  (1) Charge power constraint

  $$0 \leq P_t^{ch} \leq P_{\max}^{ch} \quad \forall t \in T \tag{20}$$

  (2) Discharge power constraint

  $$0 \leq P_t^{dch} \leq P_{\max}^{dch} \quad \forall t \in T \tag{21}$$

  (3) Cannot simultaneously charge and discharge constraints

  $$P_t^{ch} \times P_t^{dch} = 0 \quad \forall t \in T \tag{22}$$

  (4) Relationship between SOC and charging power

  $$S_{t+1} = S_t + \left( P_t^{ch} \times \eta^{ch} - P_t^{dch} \right) \times \Delta t \quad \forall t \in T \tag{23}$$

  (5) Maximum and minimum battery capacity status of EV

  $$S^{\min} \leq S_t \leq S^{\max} \quad \forall t \in T \tag{24}$$

  (6) Battery initial capacity state of EV

  $$S_{ts} = S^{ini} \tag{25}$$

  (7) Battery termination capacity status of EV

  $$S_{te} = S^{fin} \tag{26}$$

- For PV:
  (1) Power output constraint
  $$0 < P_t^{pv} < P_{\max}^{pv} \tag{27}$$

- For BS:
  (1) Charge/discharge power constraint
  $$-P_{\max}^{BS} < P_t^{BS} < P_{\max}^{BS} \tag{28}$$

  (2) Maximum and minimum battery capacity status of BS

  $$S_{BS}^{\min} \leq S_t^{BS} \leq S_{BS}^{\max} \quad \forall t \in T \tag{29}$$

- Power constraint of transformer

  $$P_t^{grid} \leq P_{\max}^{grid} \tag{30}$$

- Power balance constraints

$$P_t^{BS} + \sum_{i=1}^{n}\left(P_{i,t}^{ch} + P_{i,t}^{dch}\right) + P_t^{Building} = P_t^{PV} + P_t^{Grid} \tag{31}$$

### 2.4. Proposed Parking Lot Charging Scheduling Strategy

As shown in Figure 3, this subsection introduces the charging control strategy for the smart parking lot system, which determines the charging and discharging behavior of EVs and energy storage batteries in the parking lot, the energy flow between the parking lot and the grid, and the parking lot and the building. This strategy maximizes the local consumption of PV while reducing the charging cost of electric vehicle users as much as possible. In this strategy, the charging power of EVs and energy storage batteries mainly depends on the EPT. This strategy can enable a more reasonable distribution of energy among EVs. On the one hand, energy from PV or the grid is distributed among EVs, giving EVs and energy storage devices the opportunity to use cheaper energy for charging, so as to reduce their EPT and reach their EPT within a reasonable time frame. The ideal battery state (SOC), on the other hand, when the electric vehicle reaches the discharge condition, will provide energy for the building at a price lower than that of the grid, and while gaining certain benefits, it will make a certain contribution to the peak-shaving and valley-filling of the grid.

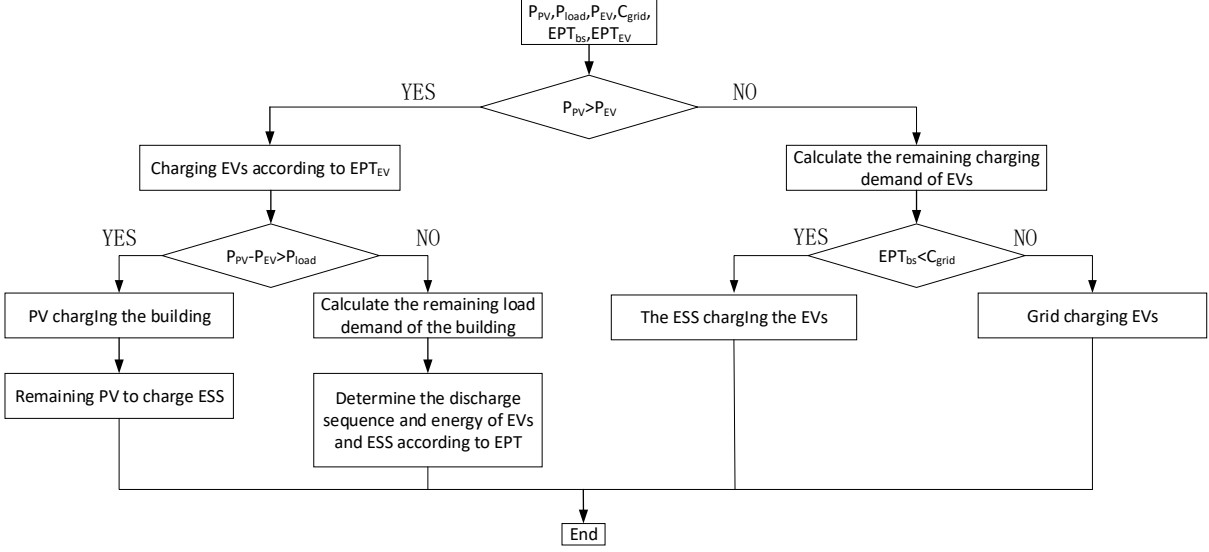

**Figure 3.** The control mechanism of the proposed control strategy.

## 3. Simulation Results and Discussion

### 3.1. Parameters Information

Table 1 shows the specific parameters of the ESS equipped with the parking lot.

**Table 1.** Parameter settings of BS.

| Variable | Charging Coefficient | Discharging Coefficient | $SOC_{min}$ | $SOC_{max}$ | $P_{ch,max}$ (kW) | $P_{dis,max}$ (kW) | E (kWh) |
|---|---|---|---|---|---|---|---|
| Value | 0.99 | 0.99 | 0.2 | 0.9 | 20 | 20 | 100 |

Figure 4 shows the 24-h load demand of the building. The time periods of 00:00–04:00 and 21:00–24:00 are power valley periods, and the electricity price is 0.3923. The time period of 05:00–08:00 is a normal period, and the electricity price is 0.6768. Meanwhile,

09:00–21:00 is a peak period, and the electricity price is 1.0436. The TOU prices of power grid and PV are shown in Figure 5.

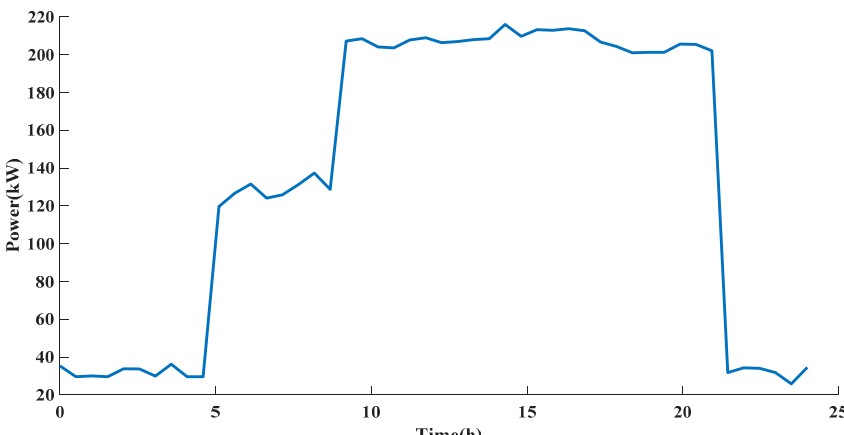

**Figure 4.** Building load demand.

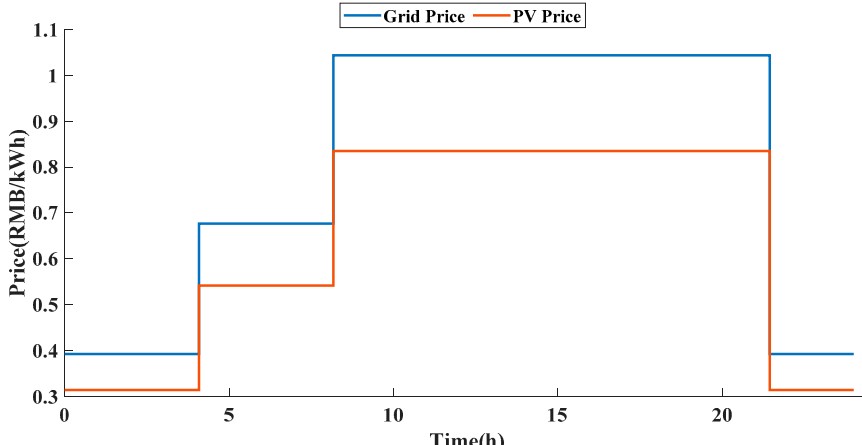

**Figure 5.** TOU Price.

The initial EPTs for EVs and ESS are randomly selected in the range of 0.5–1.0. Figure 6 shows the daily output of PV and the maximum output power of PV is 97 kW.

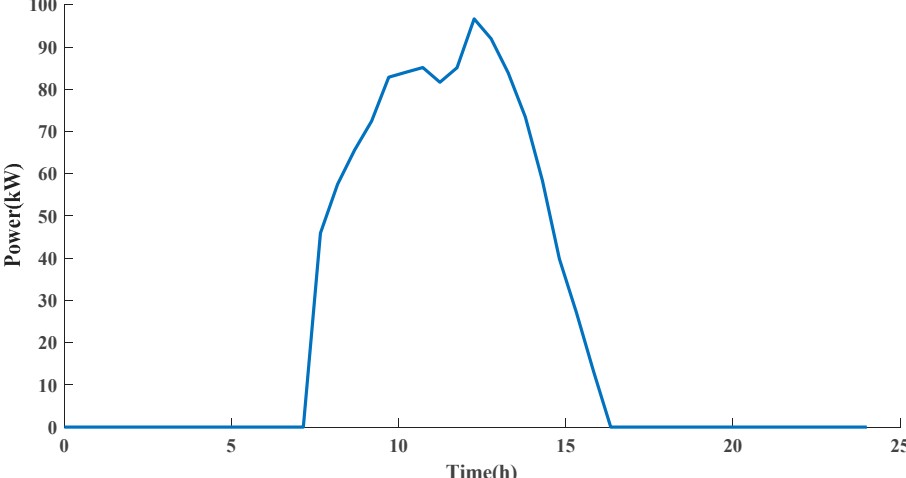

**Figure 6.** Output Power from PV.

The power of each charging pile in the parking lot is variable, and the upper limit of the power is 5kW. The relevant data of EVs are obtained from [15], including charging start time, charging end time, initial SOC, termination SOC, and battery information.

### 3.2. Case Description

According to the parameters previously set, the following four different parking energy management strategies will be simulated to observe the operation effect in a day.

Case 1: There is only a parking lot connected to the power grid. The EV adopts the immediate charging method. The charging starts at the time of arrival, and the charging process is uninterrupted until it is fully charged. It should be noted that the power grid has no restrictions on the power supply required for electric vehicle charging.

Case 2: The parking lot is equipped with ESS and PV. A one-way flow of power is maintained, only buying electricity from the grid, and the charging plan for energy storage and EVs is optimized with the minimization of operating costs as the optimization objective.

Case 3: The parking lot is equipped with ESS and PV. Using the control strategy proposed in this paper, with the minimum operating cost as the optimization goal, control of PV output, EVs charging demand, energy storage operation, and building load demand are coordinated.

Case 4: On the basis of Case 3 simulation, considering the unfavorable conditions of low PV power on cloudy days, the management strategy proposed in this paper is used to optimize ESS charging and discharging plans, EVs charging plans, and building charging plans.

### 3.3. Simulation Analysis

Figure 7 shows the EVs charging demand curve of Case 1. EVs adopt the control strategy of charging as soon as they arrive. The charging peak period of the parking lot will be concentrated when the electric vehicle just arrives at the parking lot. Since PV and ESS are not equipped, the demand load of EVs and buildings needs to be purchased from the grid. In this case, the charging cost of the EVs is 383.41, and the charging cost of the building is 5138.4.

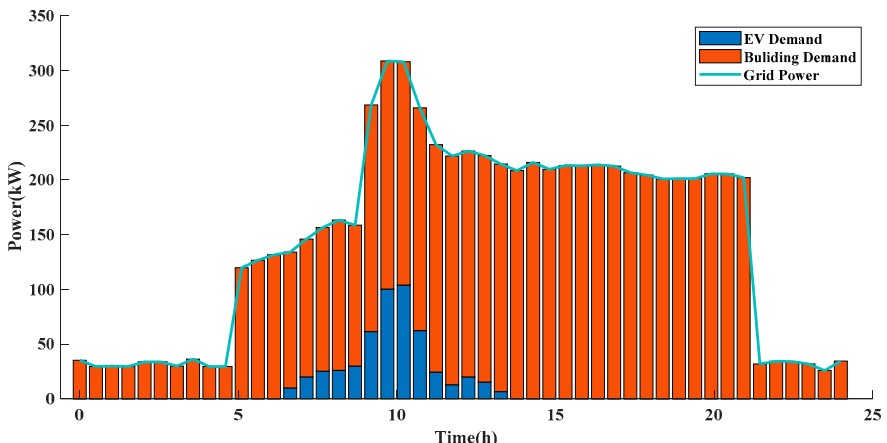

**Figure 7.** Initial EV charging demand.

Figure 8 shows the typical charging curve of EV. The EV adopts the charging method of instant charging, charging with the maximum power until reaching the user demand. Figure 9 shows the EPT variation curve of EV in the charging process, which is related to the charging power and grid price. The initial EPT of EV is 0.95, and the grid price is 0.6768. In the charging process, with the increasing proportion of low-price energy in the battery capacity, the EPT of EV is gradually decreasing. There is no charging behavior after 11:00, and the EPT remains unchanged. However, when the grid price increases to 1.0436 at 13:00, the sell price of EV also increases.

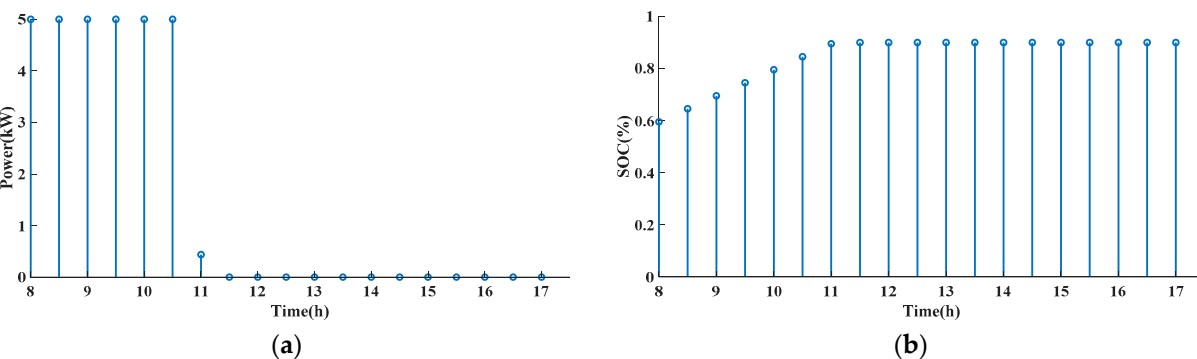

**Figure 8.** Typical charging curve of EV. (**a**) charging power; (**b**) SOC.

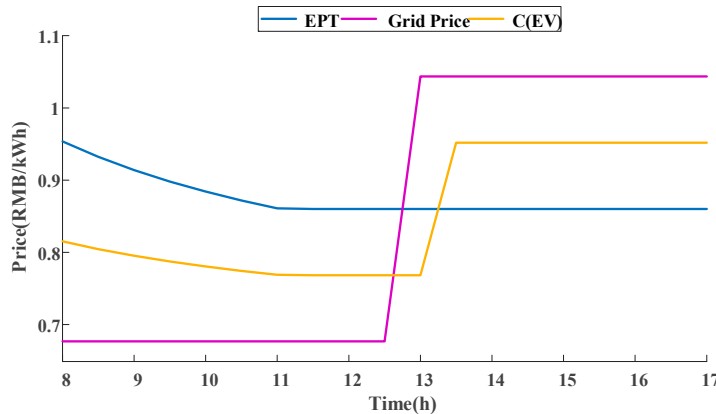

**Figure 9.** EPT and sell price of EV.

The simulation result of case 2 is shown in Figure 10. Since the power flow in this case is unidirectional, PV can only be absorbed by ESS and EVs, and the excess PV power is wasted. Compared with case 1, the charging cost of the EVs is reduced by 19.63% to 308.1701, the charging cost of the building remains unchanged, and the PV consumption rate is 46.53%.

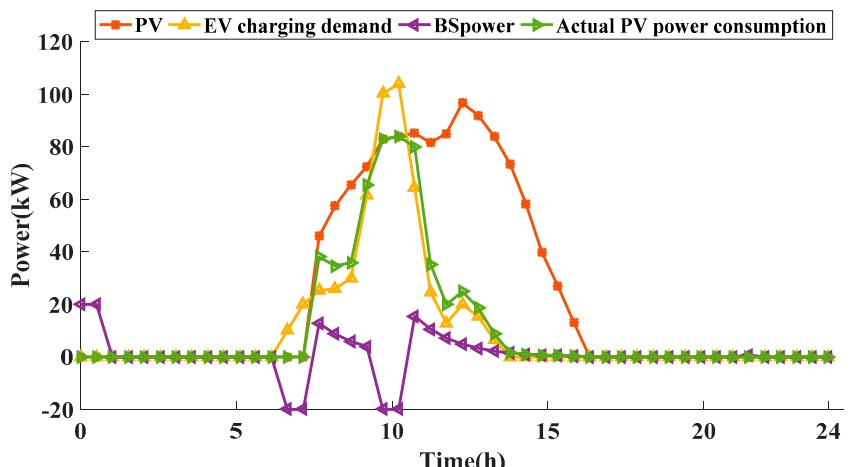

**Figure 10.** The output of each unit of the parking lot.

The simulation results of case 2 are shown in Figures 11–16. Figure 11 shows the operation of PV, ESS, and EVs in the parking lot. It can be seen from the simulation result that the charging demand of EVs has not been changed. At 6:00–7:30 and 9:30–10:30, the

PV power at this time gradually increases, but it cannot meet the charging demand of EVs. At this time, energy storage devices are needed to supplement the shortfall. At other times, PV output can fully meet the charging needs of EVs. The excess PV power is first used to charge the ESS, and if there is surplus, it is used to provide energy to the building. At this time, the charging cost of the EVs is 308.1701, a reduction of 19.63% compared to case 1.

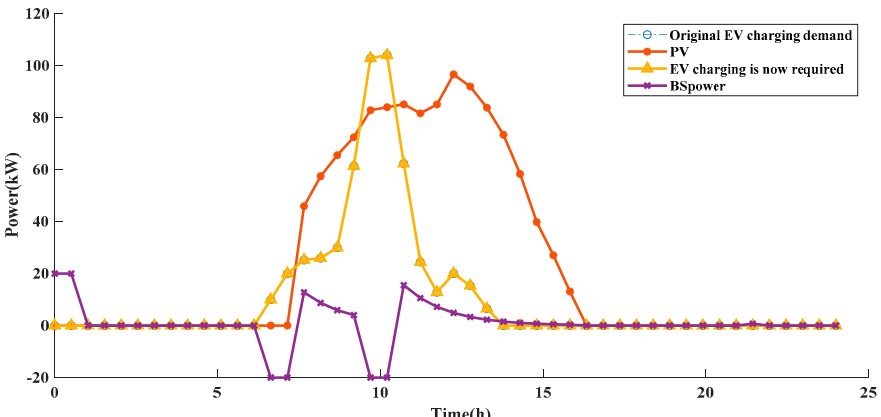

**Figure 11.** The output of each unit of the parking lot.

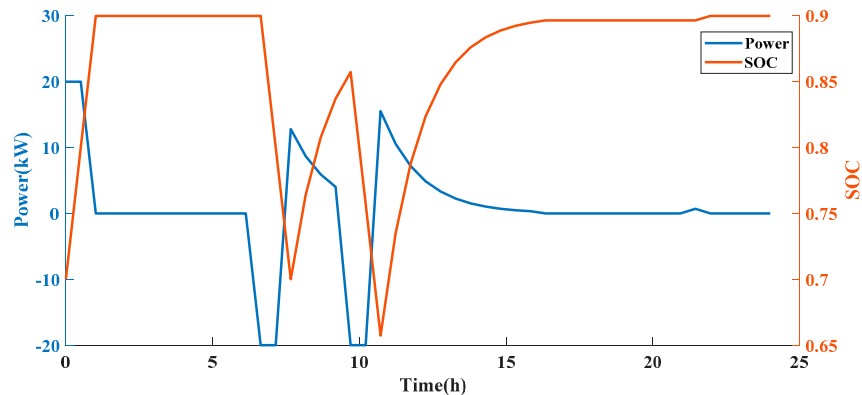

**Figure 12.** Charging Power and SOC of Energy Storage System.

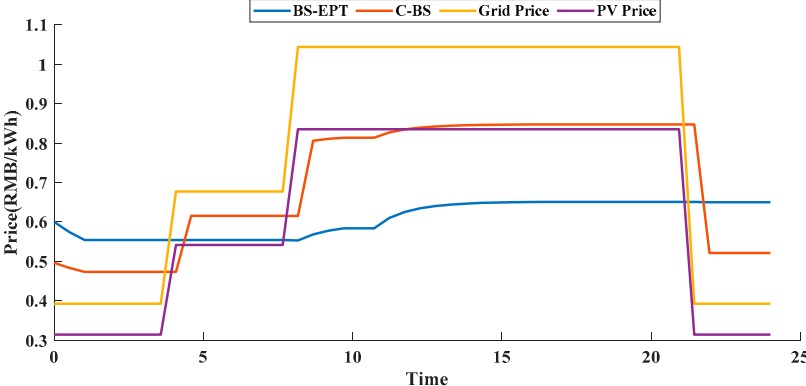

**Figure 13.** EPT and sell price of Energy Storage System.

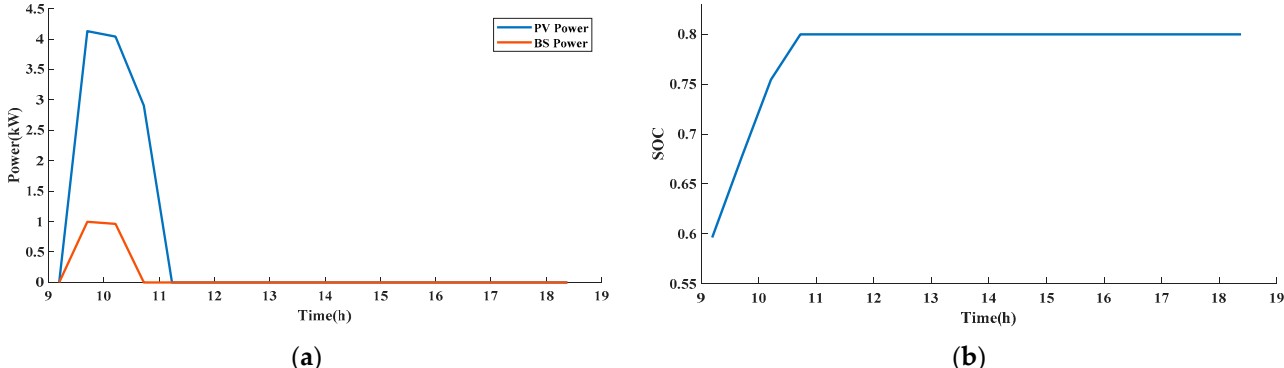

**Figure 14.** Typical charging curve of EV. (**a**) charging power; (**b**) SOC.

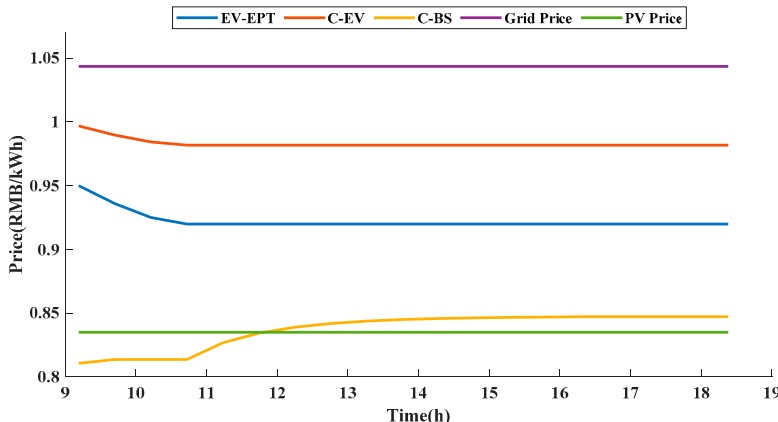

**Figure 15.** EPT and sell price of EV.

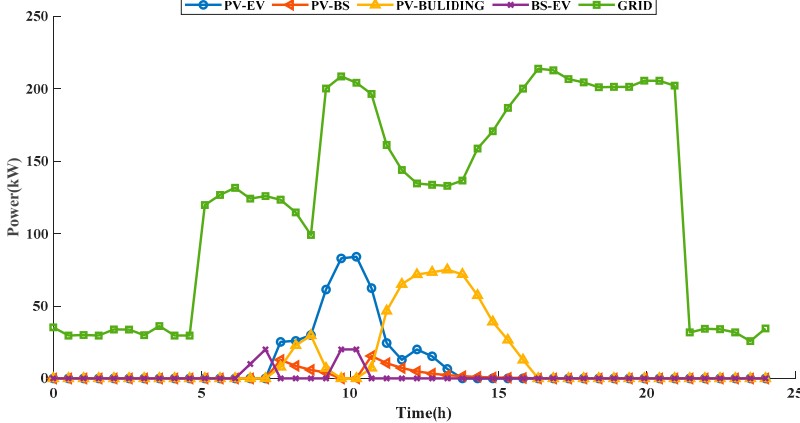

**Figure 16.** Parking lot power distribution and grid power purchase.

Figure 12 shows the charging and discharging power curve and SOC curve of the energy storage system. At 00:00, it is at a low price, and it is charged for the energy storage system by purchasing electricity from the grid. There was no output at 06:00 h, and the energy storage price was lower than the grid price, so the energy storage system provided energy for EVs. After 07:00, the PV output gradually increased. Although the price of PV may be higher than that of other energy sources, the priority of PV is the highest.

Figure 12 is the EPT curve of the energy storage system. In the initial stage of charging, the grid price is lower than the initial energy storage EPT, the proportion of low-price energy in the energy storage system is increasing, and the energy price is continuously

decreasing. Conversely, in the subsequent charging process, due to the increase of photovoltaic electricity prices and grid electricity prices, energy prices continue to increase.

An EV is extracted from the simulation results. The charging power and SOC curves are shown in Figure 13. EVs still use the charging method of instant charging, but compared with Case1, the energy composition has changed greatly. The priority is to use PV output, followed by the energy storage system, and finally to purchase electricity from the grid. Since the output of PV and energy storage in this case can meet the charging demand of EV, it is not necessary to purchase electricity from the grid.

The EPT variation curve of EV during charging is shown in Figure 14. The initial EPT is 0.95, which is higher than the PV price and energy storage price, and lower than the grid price. With the charging of PV and energy storage systems for EV, the proportion of low-price energy in batteries is increasing, and the EPT of EV is decreasing until constant.

Figure 15 shows the distribution of PV power and the purchase of electricity from the grid. From the simulation results, it can be seen that compared to case 1, the power purchase from the grid is relatively reduced, because part of the PV power is provided to the building. The energy source of the EV is mainly PV, and the energy storage is used as a backup power source to provide energy. At that time, the charging cost of the building is 5015.73, a reduction of 2.4% compared to case 1. The consumption rate of PV is 100%, which is an increase of 53.47% compared to case2.

The simulation results of case4 are shown in Figures 17 and 18. This case considers the decline in PV output on cloudy and rainy days. Compared with the simulation results of case3, in the 09:00—11:30 period, since the output of PV and ESS cannot meet the charging demand of EVs, the management strategy proposed in this paper is used to adjust the charging plan to make full use of PV, and transfer the charging time to 14:00—20:00 for charging by using the power grid, which reduces the charging cost and increases the local consumption rate of PV. The charging cost of EVs is 327.83, the charging cost of buildings is 5138.4, and the local consumption rate of PV is 100%.

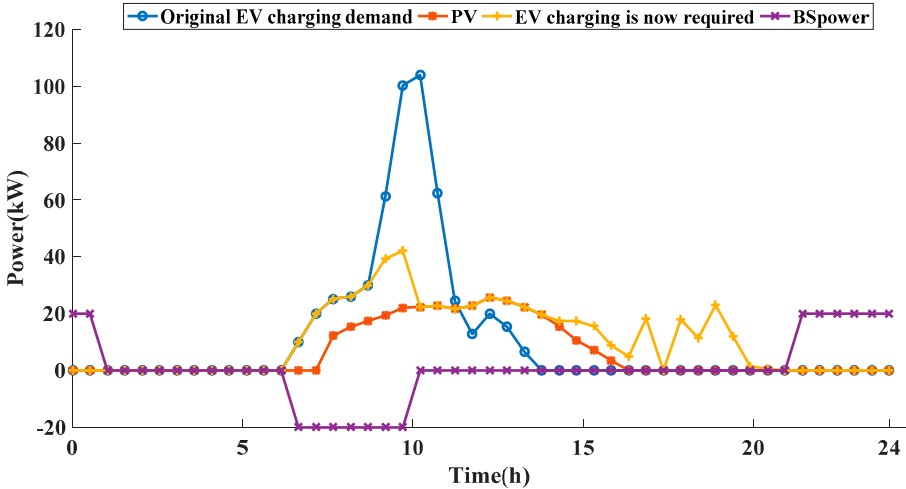

**Figure 17.** The output of each unit of the parking lot.

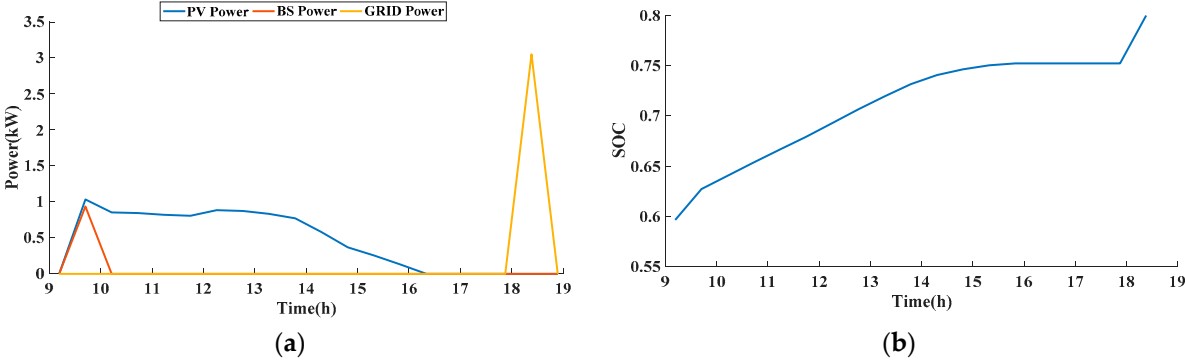

**Figure 18.** Typical charging curve of EV. (**a**) charging power; (**b**) SOC.

An EV is extracted from the simulation results. The charging power and SOC curves are shown in Figure 18. Compared with case3, the charging plan of EV has changed greatly. Firstly, the charging time and power are changed according to the PV power. Secondly, in order to reduce the charging cost, electricity is purchased from the grid at a low price.

## 4. Conclusions

This paper designs a parking lot energy management system that integrates ESS and PV systems. The charging plan of EVs, ESS, and buildings is optimized to minimize the charging cost and maximize the PV consumption, and the bidirectional flow of energy between the parking lot and the grid is realized. By comparing the simulation results of four cases, the effectiveness of the proposed strategy is proven.

Firstly, comparing the simulation results, when the power of PV and ESS cannot meet the charging demand of EVs, it is necessary to adjust the charging plan of EVs according to the PV power, and to purchase electricity from the grid at low prices. On the contrary, the charging plan of EVs remains unchanged, and the excess energy is first used to charge the ESS and then to charge the building. Secondly, compared with case1, the proposed control strategy reduces the charging cost of EVs and buildings and improves the consumption rate of PV.

**Author Contributions:** Conceptualization, Z.Y. and R.Y.; methodology, H.D. and Q.Z.; software, S.G.; investigation, T.G. and Z.Y.; resources, X.H.; writing—original draft preparation, D.M.; writing— review and editing, Z.Y.; supervision, X.H. All authors have read and agreed to the published version of the manuscript.

**Funding:** This research was funded by State Grid Corporation Science and Technology Project (Urban power grid dispatching for large-scale electric vehicle access technical study, 5108-202118041A-0-0-00).

**Institutional Review Board Statement:** Ethical review and approval were waived for this study, since it did not involve humans or animals.

**Informed Consent Statement:** The study did not involve humans.

**Data Availability Statement:** The data generated in this study cannot be shared.

**Conflicts of Interest:** The authors declare no conflict of interest. Qi Zhao and Hongen Ding are employees of the State Grid Jiangsu Electric Power Co., Ltd. The paper reflects the views of the scientists, and not the company.

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
