# Peer review of "Operation Strategy of Parking Lots Integrated with PV and Considering Energy Price Tags"

_wevj, doi:10.3390/wevj12040205_

Round 1
Reviewer 1 Report
This paper investigates an operation strategy of parking lots integrated with PV and considering energy price tags. However, there are some following problems that should be clarified or modified.:
- There are some grammar mistakes, such as line 60 and line 113.
- The simulation conditions in this paper are not complete (for example, although the power of each charging pile is variable, what is the power limit? What type of EV is used for simulation? What is the battery capacity? What is the source of the data?).
- The literature review requires improvement along with more recent papers.
- Though conclusions explained in point by point, it is still required to improve this part with a clear suggestion.
Author Response
Response to Reviewer 1 Comments
This paper investigates an operation strategy of parking lots integrated with PV and considering energy price tags. However, there are some following problems that should be clarified or modified.:
Point 1: There are some grammar mistakes, such as line 60 and line 113. 

Response 1: The sentence of the paper has been revised. The specific changes have been marked in the paper.
Point 2: The simulation conditions in this paper are not complete (for example, although the power of each charging pile is variable, what is the power limit? What type of EV is used for simulation? What is the battery capacity? What is the source of the data?)
Response 2: Data have been supplemented in the paper. The specific additions are:
The power of each charging pile in the parking lot is variable, and the upper limit of the power is 5kW. The relevant data of electric vehicles is obtained from [15], including charging start time, charging end time, initial SOC, termination SOC and battery information.
Point 3: The literature review requires improvement along with more recent papers. 

Response 3: The introduction has been revised and the recent papers has been cited. The specific changes have been marked in the paper.
Point 4: Though conclusions explained in point by point, it is still required to improve this part with a clear suggestion. 

Response 4: The conclusions has been modified. The specific content is:
This paper designs a parking lot energy management system that integrates ESS and PV systems. The charging plan of EVs, ESS and buildings is optimized to minimize the charging cost and maximize the PV consumption, and the bidirectional flow of energy between the parking lot and the grid is realized. By comparing the simulation results of four cases, the effectiveness of the proposed strategy is proved.
Firstly, comparing the simulation results, when the power of PV and ESS cannot meet the charging demand of EVs, it is necessary to adjust the charging plan of EVs according to the PV output, and purchase electricity from the grid at low prices. On the contrary, the charging plan of EVs remains unchanged, and the excess energy is first used to charge the stored energy, and then to charge the building. Secondly, compared with case1, the proposed control strategy reduces the charging cost of EVs and buildings, and improves the consumption rate of PV.

Reviewer 2 Report
This manuscript established a smart parking lot management system which combines the energy storage system (ESS) with the photovoltaic (PV) system. The topic is hot but this manuscript should be fully revised before it is accepted by a journal.
Major Issues:
Some contributions are mentioned in abstract and introduction. However, the simulation results cannot support these contributions and are not sufficient to illustrate the highlights of this research.
“short-term PV power prediction”: this manuscript uses an already known PV power data, and fails to make predictions.
“improve the utilization rate of the PV power”: in Case 1, there are no PV devices. In Case 2, all the PV energy is used. The result only shows the absorbing capacity of Base Load and BS. The advantage of the operation strategy has not been reflected.
“charging scheduling strategy”: Since the power of PV and BS is sufficient for EV charging, the scheduling strategy is not used. There should be some other cases, in which the grid power is used to charge EVs, to show the effect of scheduling strategy.
The simulation cases used in this manuscript is an easy case for the optimization of the management system, and many other simpler algorithms could also realize the same effects in these two cases. To illustrate the contributions of this method, the system established in this article should be applied to more unfavorable conditions. For example, low PV power in a cloudy day, low initial BS SOC, much more EV charging demand, etc. Therefore, the data and analysis of this research couldn't support the novelty. And an extra test is needed to verify the method.
Minor Issues:
p7.235 the unit should be indicated in the table
Author Response
Response to Reviewer 2 Comments
This manuscript established a smart parking lot management system which combines the energy storage system (ESS) with the photovoltaic (PV) system. The topic is hot but this manuscript should be fully revised before it is accepted by a journal.
Major Issues:
Point 1: Some contributions are mentioned in abstract and introduction. However, the simulation results cannot support these contributions and are not sufficient to illustrate the highlights of this research.
Response 1: The sentence of the paper has been revised and simulation cases have been added to ensure that the simulation results can support the contributions and innovations mentioned in the abstract and introduction. The specific changes have been marked in the article.
Case 2: The parking lot is equipped with ESS and PV. Maintain a one-way flow of power, only buy electricity from the grid, and optimize the charging plan for energy storage and electric vehicles with the minimization of operating costs as the optimization objective.
Case 4: On the basis of Case 3 simulation, considering the unfavorable conditions of low PV power in cloudy days, use the management strategy proposed in this paper to optimize ESS charging and discharging plans, EVs charging plans, and building charging plans.
Point 2: “short-term PV power prediction”: this manuscript uses an already known PV power data, and fails to make predictions.
Response 2: The sentence has been revised in the paper. The main work of this paper is to formulate parking lot energy management strategies based on the collected real data. The processing of PV forecast deviation will be studied in a later paper.
Point 3: “improve the utilization rate of the PV power”: in Case 1, there are no PV devices. In Case 2, all the PV energy is used. The result only shows the absorbing capacity of Base Load and BS. The advantage of the operation strategy has not been reflected.
Response 3: A new comparison case is added :“Case 2: The parking lot is equipped with ESS and PV. Maintain a one-way flow of power, only buy electricity from the grid, and optimize the charging plan for energy storage and electric vehicles with the minimization of operating costs as the optimization objective”. By comparing the unidirectional / bidirectional simulation results of energy flow between the parking lot and the grid, it is proved that the strategy proposed in this paper can improve the consumption rate of PV power. The specific changes have been marked in the paper.
Point 4: “charging scheduling strategy”: Since the power of PV and BS is sufficient for EV charging, the scheduling strategy is not used. There should be some other cases, in which the grid power is used to charge EVs, to show the effect of scheduling strategy.
Response 4: A new comparison case is added :“Case 4: On the basis of Case 3 simulation, considering the unfavorable conditions of low PV power in cloudy days, use the management strategy proposed in this paper to optimize ESS charging and discharging plans, EVs charging plans, and building charging plans”. Considering that the decline in PV power cannot meet the charging demand of EVs on cloudy and rainy days, compared with the simulation results of other cases, the charging cost and PV consumption rate have been improved to varying degrees, which proves the effectiveness of the method proposed in this paper. The specific changes have been marked in the paper.
Point 5: The simulation cases used in this manuscript is an easy case for the optimization of the management system, and many other simpler algorithms could also realize the same effects in these two cases. To illustrate the contributions of this method, the system established in this article should be applied to more unfavorable conditions. For example, low PV power in a cloudy day, low initial BS SOC, much more EV charging demand, etc. Therefore, the data and analysis of this research couldn't support the novelty. And an extra test is needed to verify the method.
Response 5: Two new comparison cases are added :“Case 2: The parking lot is equipped with ESS and PV. Maintain a one-way flow of power, only buy electricity from the grid, and optimize the charging plan for energy storage and electric vehicles with the minimization of operating costs as the optimization objective” and “Case 4: On the basis of Case 3 simulation, considering the unfavorable conditions of low PV power in cloudy days, use the management strategy proposed in this paper to optimize ESS charging and discharging plans, EVs charging plans, and building charging plans”. The simulation results of case1 and case2 can prove that the parking lot system integrated with PV and ESS can reduce charging costs; The simulation results of case2 and case3 show that the PV consumption rate can be improved and the profit of the parking lot can be further increased when the energy of the parking lot and the power grid has bidirectional flow; The simulation results of case3 and case4 can prove that the method proposed in this paper is equally effective in the face of unfavorable conditions and has good adaptability.
Minor Issues:
Point 6: p7.235 the unit should be indicated in the table
Response 6: Modified and marked in the paper.

Round 2
Reviewer 1 Report
Accept in present form
Reviewer 2 Report
The authors have stressed most of the comments in the first round. Thank you for the revision. The manuscript has improved a lot, and I think it is ready for publication.